# Synthesis of Novel Benzenesulfonamide-Bearing Functionalized Imidazole Derivatives as Novel Candidates Targeting Multidrug-Resistant *Mycobacterium abscessus* Complex

**DOI:** 10.3390/microorganisms11040935

**Published:** 2023-04-03

**Authors:** Benas Balandis, Povilas Kavaliauskas, Birutė Grybaitė, Vidmantas Petraitis, Rūta Petraitienė, Ethan Naing, Andrew Garcia, Ramunė Grigalevičiūtė, Vytautas Mickevičius

**Affiliations:** 1Department of Organic Chemistry, Kaunas University of Technology, Radvilėnų Rd. 19, LT-50254 Kaunas, Lithuania; benas.balandis@ktu.lt (B.B.); birute.grybaite@ktu.lt (B.G.); vytautas.mickevicius@ktu.lt (V.M.); 2Transplantation-Oncology Infectious Diseases Program, Division of Infectious Diseases, Department of Medicine, Weill Cornell Medicine of Cornell University, 1300 York Ave., New York, NY 10065, USA; 3Institute for Genome Sciences, School of Medicine, University of Maryland Baltimore, 655 W. Baltimore Street, Baltimore, MD 21201, USA; 4Institute of Infectious Diseases and Pathogenic Microbiology, Birštono Str. 38A, LT-59116 Prienai, Lithuania; 5Biological Research Center, Lithuanian University of Health Sciences, Tilžės Str. 18/7, LT-47181 Kaunas, Lithuania

**Keywords:** benzenesulfonamides, imidazoles, S-alkylated, antimycobacterial activity, antifungal activity

## Abstract

Infections caused by drug-resistant (DR) *Mycobacterium abscessus* (*M. abscessus*) complex (MAC) are an important public health concern, particularly when affecting individuals with various immunodeficiencies or chronic pulmonary diseases. Rapidly growing antimicrobial resistance among MAC urges us to develop novel antimicrobial candidates for future optimization. Therefore, we have designed and synthesized benzenesulfonamide-bearing functionalized imidazole or *S-*alkylated derivatives and evaluated their antimicrobial activity using multidrug-resistant *M. abscessus* strains and compared their antimycobacterial activity using *M. bovis* BCG and *M. tuberculosis* H37Ra. Benzenesulfonamide-bearing imidazole-2-thiol compound **13,** containing 4-CF_3_ substituent in benzene ring, showed strong antimicrobial activity against the tested mycobacterial strains and was more active than some antibiotics used as a reference. Furthermore, an imidazole-bearing 4-F substituent and S-methyl group demonstrated good antimicrobial activity against *M. abscessus* complex strains, as well as *M. bovis* BCG and *M. tuberculosis* H37Ra. In summary, these results demonstrated that novel benzenesulfonamide derivatives, bearing substituted imidazoles, could be further explored as potential candidates for the further hit-to-lead optimization of novel antimycobacterial compounds.

## 1. Introduction

Infections caused by nontuberculous mycobacteria (NTM) remain a challenging and emerging public health threat, particularly in individuals undergoing chemotherapy or patients with underlying lung conditions [1]. The incidence of infections caused by NTM is increasing globally, and it can be challenging to diagnose and treat due to the diverse range of NTM species and varying patterns of antibiotic susceptibility. In addition, some NTM developed resistance to multiple antibiotics, making the infections caused by NTM difficult to treat and worsening the treatment prognosis [2,3,4]. Therefore, it is critical to develop novel small molecule antimicrobial candidates targeting NTM in particularly multidrug-resistant (MDR) strains. 

*Mycobacterium abscessus* (*M. abscessus*) complex (MABC) is responsible for the majority of MDR NTM infections. MABC, consisting of genetically and phylogenetically related subspecies *M. abscessus* subsp. *abscessus, massiliense*, and *bolletii,* which often harbor multiple instinctive and acquired antimicrobial resistance determinants, as well as numerous virulence factors, make *M. abscessus* a clinically important pathogen [5]. Treatment or management of infections caused by *M. abscessus* requires long-term-to-lifetime treatment using multiple antibiotics. The treatment regime and duration is highly dependent on susceptibility of *M. abscessus* to macrolides [6]. Macrolide-susceptible *M. abscessus* infections require treatment using combinations of parenteral and inhalable antibiotics, which follows a maintenance period with at least three different inhalable and/or systemic antibiotics [7]. Macrolide-resistant MABC infections require a prolonged treatment duration, as well as an increased number of antibiotics, consequently making these infections extremely challenging. Therefore, it is crucial to discover novel candidates targeting MDR NTM with increased focus on *M. abscessus* [8]. 

Benzenesulfonamide scaffold have been widely used as a potent pharmacophore in medicinal chemistry [9,10]. Benzenesulfonamide nucleus-containing derivatives are widely explored in medicinal chemistry due to their ability to modulate various biological targets [11,12,13,14,15,16]. The sulfonamide group in the scaffold can act as a hydrogen bond acceptor and donor, providing a versatile platform for molecular modifications and improving the pharmacokinetic and pharmacodynamic properties of the targeted molecule. The benzene ring, on the other hand, is hydrophobic and contributes to the lipophilicity of the molecule. This can enhance the biologically active compound’s ability to penetrate cell membranes or lipid layers that are found on the cell wall of various mycobacteria. Moreover, benzenesulfonamide derivatives have been previously reported to be good inhibitors of carbonic anhydrases (CAs) [17,18,19] and therefore exhibiting antimicrobial or anticancer activity. Moreover, CAs in *M. abscessus* have been previously reported to be a promising antimicrobial target since the pharmacological or molecular inhibition of CAs in *M. abscessus* leads to defective growth or virulence [20,21,22,23]. Therefore, benzenesulfonamide nucleus-bearing compounds could be attractive scaffolds for antimicrobial discovery targeting NTM. 

Imidazole is a common heterocyclic scaffold that is often found in many natural and synthetic bioactive compounds and approved drugs. Molecules containing imidazole moiety can possess a wide variety of biological properties [24,25,26,27,28,29,30,31]. Structurally, imidazole can be further chemically modified with numerous substituents, making imidazole an extremely versatile scaffold [32,33,34]. Moreover, substituting a hydrophobic group at position 2 of the imidazole ring can improve the compound’s lipophilicity, which can increase its membrane permeability and enhance the activity of antimicrobial compounds bearing an imidazole nucleus against mycobacteria.

In this study, we have synthesized novel benzenesulfonamide moiety-bearing functionalized imidazole derivatives containing various S-alkyl substituents and evaluated their antimicrobial activity against drug-resistant *M. abscessus* complex strains. 

## 2. Materials and Methods

### 2.1. Reagents and Equipment Used for Synthesis and Characterization of Compounds

Reagents were purchased from Sigma-Aldrich (St. Louis, MO, USA). The melting points were determined on a MEL-TEMP (Electrothermal, Bibby Scientific Company, Burlington, NJ, USA) melting point apparatus and were uncorrected. IR spectra (ν, cm^−1^) were recorded on a Perkin–Elmer Spectrum BX FT–IR spectrometer using KBr pellets. The ^1^H and ^13^C NMR spectra were recorded in DMSO-*d*_6_ medium on Brucker Avance III (400, 101 MHz) spectrometer. Chemical shifts (δ) are reported in parts per million (ppm) calibrated from TMS (0 ppm) as an internal standard for ^1^H NMR, and DMSO-*d*_6_ (39.43 ppm) for ^13^C NMR. Elemental analysis was performed on a CE-440 elemental analyzer (Exeter Analytical Inc., North Chelmsford, MA, USA). The reaction course and purity of the synthesized compounds were monitored by TLC using aluminium plates precoated with silica gel 60 F_254_ (MerckKGaA, Darmstadt, Germany).

### 2.2. Synthesis

Complete synthesis of compounds **2**, **4**, **5**, **7**, and **8**, as well as compounds **9**, **11**, **12**, **14, 15, 18b** was described in our previous study [35] and were resynthesized accordingly in this study. These compounds were further used for S-alkylation reactions. All of spectra data on compounds **2**, **4**, **5**, **7**, and **8** as well as compounds **9**, **11**, **12**, **14**, **15**, **18b** was described in our previous publication [35].


**General procedure for the synthesis of compounds 3 and 6.**


Amine **1** (1.72 g, 10 mmol) was dissolved in boiling water (40 mL). Then, the solution of corresponding α-haloketone (12 mmol) in 10 mL of 1,4-dioxane was added dropwise to the mixture. The reaction mixture was heated at reflux for 2 h, then it was cooled down and the precipitate was filtered off, washed with diethyl ether, and recrystallized from 1,4-dioxane to afford compounds **3** and **6**. 

**3-((2-(4-bromophenyl)-2-oxoethyl)amino)benzenesulfonamide** (**3**). White solid, yield 3.12 g (85%); m.p. 236–237 °C; IR (KBr) (v, cm^−1^): 3382, 3262, 1692; ^1^H NMR (400 MHz, DMSO-d_6_) δ (ppm): 4.72 (s, 2H, CH_2_), 6.40 (br. s, 1H, NH), 6.84 (dd, 1H, J = 8.1, 2.3, H_ar_), 7.01 (d, 1H, J = 7.6, H_ar_), 7.11 (br. s, 1H, H_ar_), 7.18 (br. s, 2H, NH_2_), 7.23 (t, 1H, J = 7.9 Hz, H_ar_), 7.79 (d, 2H, J = 8.1 Hz, H_ar_), 8.00 (d, 2H, J = 8.1 Hz, H_ar_); ^13^C NMR (101 MHz, DMSO-d_6_) (δ, ppm): 49.96, 109.00, 112.96, 115.28, 127.68, 129.23, 129.93, 131.87, 134.04, 144.75, 148.49, 195.66; Anal. Calcd. for C_14_H_13_BrN_2_O_3_S: C 45.54; H 3.55; N 7.59 %. Found: C 45.57; H 3.55; N 7.54 %.

**3-((2-oxo-2-(4-(trifluoromethyl)phenyl)ethyl)amino)benzenesulfonamide** (**6**). White solid, yield 2.97 g (83%); m.p. 228–229 °C; IR (KBr) (v, cm^−1^): 3390, 3268, 1698; ^1^H NMR (400 MHz, DMSO-d_6_) δ (ppm): 4.81(s, 2H, CH_2_), 6.44 (br. s, 1H, NH), 6.84 (d, 1H, J = 7.7, H_ar_), 7.03 (d, 1H, J = 7.7, H_ar_), 7.13 (br. s, 1H, H_ar_), 7.19 (br. s, 2H, NH_2_), 7.24 (t, 1H, J = 7.9 Hz, H_ar_), 7.95 (d, 2H, J = 8.0 Hz, H_ar_), 8.26 (d, 2H, J = 8.0 Hz, H_ar_); ^13^C NMR (101 MHz, DMSO-d_6_) (δ, ppm): 50.03, 109.06, 113.05, 115.29, 122.42, 125.13, 125.73, 125.77, 125.80, 125.84, 128.78, 129.26, 138.28, 144.78, 148.47, 196.02; Anal. Calcd. for C_15_H_13_F_3_N_2_O_3_S: C 50.28; H 3.66; N 7.82 %. Found: C 50.28; H 3.62; N 7.80 %.


**General procedure for the synthesis of imidazoles 10 and 13.**


An amount of 2 mmol of compounds **3**, **6** was dissolved in the solution of glacial acetic acid (5 mL) and HCl (1 mL), and KSCN (0.78 g, 8 mmol) was added. The reaction mixture was heated at reflux for 4 h, then it was cooled down, diluted with water, and the precipitate was filtered off and washed with water and n-hexane.

**3-(4-(4-bromophenyl)-2-thioxo-2,3-dihydro-1H-imidazol-1-yl)benzenesulfonamide** (**10**). Yellowish solid, yield 0.59 g (72%); m.p. 280–281 °C; IR (KBr) (v, cm^−1^): 3258, 2729, 1485; ^1^H NMR (400 MHz, DMSO-d_6_) δ (ppm): 7.52 (s, 2H, NH_2_), 7.60–7.79 (m, 5H, H_ar_), 7.84–7.98 (m, 2H, H_ar_), 8.02 (s, 1H, H_ar_), 8.18 (s, 1H, CH), 13.12 (s, 1H, SH); ^13^C NMR (101 MHz, DMSO-d_6_) (δ, ppm): 116.38, 120.95, 122.98, 125.01, 126.26, 126.86, 127.65, 129.25, 129.68, 131.81, 137.79, 144.76, 163.17; Anal. Calcd. For C_15_H_12_BrN_3_O_2_S_2_: C 43.91; H 2.95; N 10.24 %. Found: C 43.99; H 2.91; N 10.21 %.

**3-(2-thioxo-4-(4-(trifluoromethyl)phenyl)-2,3-dihydro-1H-imidazol-1-yl)benzenesulfonamide** (**13**). Light brown solid, yield 0.51 g (75%); m.p. 226–227 °C; IR (KBr) (v, cm^−1^): 3141, 2737, 1489; ^1^H NMR (400 MHz, DMSO-d_6_) δ (ppm): 7.53 (s, 2H, NH_2_), 7.72–8.05 (m, 7H, H_ar_), 8.13–8.24 (m, 2H, CH, H_ar_), 13.26 (s, 1H, SH); ^13^C NMR (101 MHz, DMSO-d_6_) (δ, ppm): 117.77, 123.06, 124.75, 125.14, 125.91, 125.95, 127.24, 127.67, 127.99, 129.32, 129.71, 131.53, 137.70, 144.80, 163.62; Anal. Calcd. For C_16_H_12_F_3_N_3_O_2_S_2_: C 48.12; H 3.03; N 10.52; %. Found: C 48.10; H 2.99; N 10.47 %.


**General procedure for the synthesis of S-alkylated compounds 16–22a-c.**


Imidazole **9–15** (1.0 mmol) was dissolved in DMF (3 mL). Triethylamine (0.5 mL) and corresponding alkyl halide (1.5 mmol) were added dropwise, and the reaction mixture was stirred at room temperature for 2–3 h. Then, the reaction mixture was diluted with 20 mL of water. The precipitate was filtered off, washed with water and diethyl ether, dried, and recrystallized from propan-2-ol.

**3-(2-(methylthio)-4-phenyl-1H-imidazol-1-yl)benzenesulfonamide** (**16a**). White solid, yield 0.29 g (84%); m.p. 148–149 °C; IR (KBr) (v, cm^−1^): 3327, 1483; ^1^H NMR (400 MHz, DMSO-d_6_) δ (ppm): 2.63 (s, 3H, CH_3_), 7.25 (t, 1H, J = 7.4 Hz, H_ar_), 7.23 (t, 2H, J = 7.6 Hz, H_ar_), 7.57 (br. S, 2H, NH_2_), 7.75–7.98 (m, 6H, H_ar_), 8.08 (s, 1H, CH); ^13^C NMR (101 MHz, DMSO-d_6_) (δ, ppm): 14.83, 27.56, 115.34, 115.56, 118.69, 122.32, 125.44, 126.21, 126.29, 128.63, 130.00, 130.03, 130.47, 136.94, 140.57, 141.88, 145.36; Anal. Calcd. For C_16_H_15_N_3_O_3_S_2_: C 55.63; H 4.38; N 12.16; %. Found: C 55.57; H 4.35; N 12.13 %.

**3-(2-(ethylthio)-4-phenyl-1H-imidazol-1-yl)benzenesulfonamide** (**16b**). Light yellow solid, yield 0.27 g (75%); m.p. 160–161 °C; IR (KBr) (v, cm^−1^): 3324, 1482; ^1^H NMR (400 MHz, DMSO-d_6_) δ (ppm): 1.30 (t, 3H, J = 7.3 Hz, CH_3_), 3.14 (q, 2H, J = 7.3 Hz, CH_2_), 7.23 (t, 2H, J = 8.7 Hz, H_ar_), 7.57 (br. S, 2H, NH_2_), 7.70–8.00 (m, 6H, H_ar_), 8.08 (s, 1H, CH); ^13^C NMR (101 MHz, DMSO-d_6_) (δ, ppm): 14.84, 27.59, 118.84, 122.34, 124.38, 125.40, 126.87, 128.59, 128.64, 130.46, 133.43, 136.99, 141.46, 141.79, 145.35; Anal. Calcd. For C_17_H_17_N_3_O_2_S_2_: C 56.80; H 4.77; N 11.69; %. Found: C 56,79; H 4.75; N 11.64 %.

**3-(4-phenyl-2-(propylthio)-1H-imidazol-1-yl)benzenesulfonamide** (**16c**). White solid, yield 0.30 g (81%); m.p. 130–131 °C; IR (KBr) (v, cm^−1^): 3332, 1483; ^1^H NMR (400 MHz, DMSO-d_6_) δ (ppm): 0.92 (t, 3H, J = 7.3 Hz, CH_3_), 1.67 (extette, 2H, J = 7.3 Hz, CH_2_), 3.12 (t, 2H, J = 7.1 Hz, SCH_2_), 7.25 (t, 1H, J = 7.4 Hz, H_ar_), 7.40 (t, 2H, J = 7.6 Hz, H_ar_), 7.57 (br. S, 2H, NH_2_), 7.73–7.87 (m, 4H, H_ar_), 7.94 (br. S, 2H, H_ar_), 8.08 (s, 1H, CH); ^13^C NMR (101 MHz, DMSO-d_6_) (δ, ppm): 13.04, 22.41, 35.18, 118.85, 122.36, 124.36, 125.40, 126.86, 128.58, 128.65, 130.45, 133.43, 137.00, 141.40, 141.92, 145.36; Anal. Calcd. For C_18_H_19_N_3_O_2_S_2_: C 57.89; H 5.13; N 11.25; %. Found: C 57.88; H 5.09; N 11.29 %.

**3-(4-(4-bromophenyl)-2-(methylthio)-1H-imidazol-1-yl)benzenesulfonamide** (**17a**). White solid, yield 0.36 g (85%); m.p. 168–169 °C; IR (KBr) (v, cm^−1^): 3351, 1480; ^1^H NMR (400 MHz, DMSO-d_6_) δ (ppm): 2.63 (s, 3H, CH_3_), 7.54–7.62 (m, 4H, NH_2_, H_ar_), 7.76–7.84 (m, 4H, H_ar_), 7.94 (br. S, 2H, H_ar_), 8.16 (s, 1H, CH); ^13^C NMR (101 MHz, DMSO-d_6_) (δ, ppm): 15.44, 119.39, 119.60, 122.11, 125.52, 126.32, 128.42, 130.58, 131.50, 132.70, 136.77, 140.23, 143.35, 145.45; Anal. Calcd. For C_16_H_14_BrN_3_O_2_S_2_: C 45.29; H 3.33; N 9.90; %. Found: C 45.25; H 3.29; N 9.90 %.

**3-(4-(4-bromophenyl)-2-(ethylthio)-1H-imidazol-1-yl)benzenesulfonamide** (**17b**). White solid, yield 0.38 g (80%); m.p. 142–143 °C; IR (KBr) (v, cm^−1^): 3350, 1479; ^1^H NMR (400 MHz, DMSO-d_6_) δ (ppm): 1.30 (t, 3H, J = 7.3 Hz, CH_3_), 3.15 (q, 2H, J = 7.3 Hz, CH_2_), 7.53–7.61 (m, 4H, NH_2_, H_ar_), 7.73–7.83 (m, 4H, H_ar_), 7.90–7.96 (m, 2H, H_ar_), 8.16 (s, 1H, CH); ^13^C NMR (101 MHz, DMSO-d_6_) (δ, ppm): 14.82, 27.50, 119.42, 119.63, 122.32, 125.50, 126.34, 128.64, 130.49, 131.52, 132.72, 136.86, 140.31, 142.20, 145.35; Anal. Calcd. For C_17_H_16_BrN_3_O_2_S_2_: C 46.58; H 3.68; N 9.59; %. Found: C 46.61; H 3.69; N 9.55 %.

**3-(4-(4-bromophenyl)-2-(propylthio)-1H-imidazol-1-yl)benzenesulfonamide** (**17c**). White solid, yield 0.39 g (75%); m.p. 163–164 °C; IR (KBr) (v, cm^−1^): 3352, 1479; ^1^H NMR (400 MHz, DMSO-d_6_) δ (ppm): 0.92 (t, 3H, J = 7.3 Hz, CH_3_), 1.66 (extette, 2H, J = 7.2 Hz, CH_2_), 3.13 (t, 2H, J = 7.1 Hz, SCH_2_), 7.52–7.63 (m, 4H, NH_2_, H_ar_), 7.73–7.84 (m, 4H, H_ar_), 7.89–7.96 (m, 2H, H_ar_), 8.15 (s, 1H, CH); ^13^C NMR (101 MHz, DMSO-d_6_) (δ, ppm): 13.04, 22.37, 35.10, 119.42, 119.63, 122.35, 125.50, 126.32, 128.65, 130.48, 131.52, 132.71, 136.87, 140.25, 142.34, 145.38; Anal. Calcd. For C_18_H_18_BrN_3_O_2_S_2_: C 47.79; H 4.01; N 9.29; %. Found: C 47.76; H 3.98; N 9.25 %.

**3-(4-(4-chlorophenyl)-2-(methylthio)-1H-imidazol-1-yl)benzenesulfonamide** (**18a**). Brownish solid, yield 0.28 g (74%); m.p. 171–172 °C; IR (KBr) (v, cm^−1^): 3347, 1482; ^1^H NMR (400 MHz, DMSO-d_6_) δ (ppm): 2.63 (s, 3H, CH_3_), 7.45 (d, 2H, J = 8.2 Hz, H_ar_), 7.57 (br. S, 2H, NH_2_), 7.74–7.90 (m, 4H, H_ar_), 7.95 (br. S, 2H, H_ar_), 8.14 (s, 1H, CH); ^13^C NMR (101 MHz, DMSO-d_6_) (δ, ppm): 15.45, 119.33, 122.12, 125.52, 126.00, 128.42, 128.61, 130.59, 131.12, 132.35, 136.79, 140.22, 143.33, 145.46; Anal. Calcd. For C_16_H_14_ClN_3_O_2_S_2_: C 50.59; H 3.71; N 11.06; %. Found: C 50.58; H 3.68; N 11.01 %.

**3-(4-(4-chlorophenyl)-2-(propylthio)-1H-imidazol-1-yl)benzenesulfonamide** (**18c**). Light brown solid, yield 0.35 g (85%); m.p. 166–167 °C; IR (KBr) (v, cm^−1^): 3348, 1481; ^1^H NMR (400 MHz, DMSO-d_6_) δ (ppm): 0.93 (t, 3H, J = 7.3 Hz, CH_3_), 1.67 (sextet, 2H, J = 7.3 Hz, CH_2_), 3.13 (t, 2H, J = 7.1 Hz, SCH_2_), 7.46 (d, 2H, J = 8.3 Hz, H_ar_), 7.58 (br. S, 2H, NH_2_), 7.74–7.99 (m, 6H, H_ar_), 8.15 (s, 1H, CH); ^13^C NMR (101 MHz, DMSO-d_6_) (δ, ppm): 13.04, 22.38, 35.10, 119.37, 122.35, 125.50, 126.00, 128.62, 128.65, 130.48, 131.13, 132.36, 136.88, 140.23, 142.31, 145.38; Anal. Calcd. For C_18_H_18_ClN_3_O_2_S_2_: C 53.00; H 4.45; N 10.30; %. Found: C 52.99; H 4.39; N 10.30 %.

**3-(4-(4-fluorophenyl)-2-(methylthio)-1H-imidazol-1-yl)benzenesulfonamide** (**19a**). Yellowish solid, yield 0.29 g (81%); m.p. 162–163 °C; IR (KBr) (v, cm^−1^): 3341, 1484; ^1^H NMR (400 MHz, DMSO-d_6_) δ (ppm): 2.63 (s, 3H, CH_3_), 7.23 (t, 2H, J = 8.7 Hz, H_ar_), 7.57 (br. S, 2H, NH_2_), 7.75–7.98 (m, 6H, H_ar_), 8.08 (s, 1H, CH); ^13^C NMR (101 MHz, DMSO-d_6_) (δ, ppm): 15.50, 115.33, 115.55, 118.64, 122.10, 125.45, 126.20, 126.28, 128.40, 130.00, 130.03, 130.57, 136.86, 140.51, 143.04, 145.45; Anal. Calcd. For C_16_H_14_FN_3_O_2_S_2_: C 52.88; H 3.88; N 11.56; %. Found: C 52.85; H 3.89; N 11.51 %.

**3-(2-(ethylthio)-4-(4-fluorophenyl)-1H-imidazol-1-yl)benzenesulfonamide** (**19b**). Yellowish solid, yield 0.31 g (82%); m.p. 160–161 °C; IR (KBr) (v, cm^−1^): 3350, 1483; ^1^H NMR (400 MHz, DMSO-d_6_) δ (ppm): 1.30 (t, 3H, J = 7.3 Hz, CH_3_), 3.14 (q, 2H, J = 7.3 Hz, CH_2_), 7.24 (t, 2H, J = 8.7 Hz, H_ar_), 7.57 (br. S, 2H, NH_2_), 7.69–7.99 (m, 6H, H_ar_), 8.09 (s, 1H, CH); ^13^C NMR (101 MHz, DMSO-d_6_) (δ, ppm): 14.84, 27.59, 115.36, 115.57, 118.72, 122.33, 125.46, 126.22, 126.30, 128.64, 129.97, 130.48, 136.92, 140.51, 141.88, 145.37; Anal. Calcd. For C_17_H_16_FN_3_O_2_S_2_: C 54.10; H 4.27; N 11.13; %. Found: C 54.05; H 4.23; N 11.09 %.

**3-(4-(4-fluorophenyl)-2-(propylthio)-1H-imidazol-1-yl)benzenesulfonamide** (**19c**). White solid, yield 0.30 g (77%); m.p. 142–143 °C; IR (KBr) (v, cm^−1^): 3350, 1484; ^1^H NMR (400 MHz, DMSO-d_6_) δ (ppm): 0.92 (t, 3H, J = 7.3 Hz, CH_3_), 1.66 (sextet, 2H, J = 7.2 Hz, CH_2_), 3.12 (t, 2H, J = 7.1 Hz, SCH_2_), 7.23 (t, 2H, J = 8.6 Hz, H_ar_), 7.57 (br. S, 2H, NH_2_), 7.72–7.96 (m, 6H, H_ar_), 8.07 (s, 1H, CH); ^13^C NMR (101 MHz, DMSO-d_6_) (δ, ppm): 13.04, 22.40, 35.15, 115.35, 115.56, 118.70, 122.33, 125.43, 126.19, 126.27, 128.64, 130.00, 130.03, 130.47, 136.95, 140.51, 142.01, 145.36; Anal. Calcd. For C_18_H_18_FN_3_O_2_S_2_: C 55.23; H 4.63; N 10.73; %. Found: C 55.19; H 4.57; N 10.76 %.

**3-(2-(methylthio)-4-(4-(trifluoromethyl)phenyl)-1H-imidazol-1-yl)benzenesulfonamide** (**20a**). Yellow solid, yield 0.34 g (83%); m.p. 206–207 °C; IR (KBr) (v, cm^−1^): 3341, 1484; ^1^H NMR (400 MHz, DMSO-d_6_) δ (ppm): 2.65 (s, 3H, CH_3_), 7.58 (br. S, 2H, NH_2_), 7.71–7.84 (m, 4H, H_ar_), 7.91–8.12 (m, 4H, H_ar_), 8.30 (s, 1H, CH); ^13^C NMR (101 MHz, DMSO-d_6_) (δ, ppm): 15.41, 120.65, 122.18, 123.11, 124.70, 125.53, 125.57, 125.61, 125.66, 125.81, 126.72, 127.03, 128.48, 130.62, 136.68, 137.39, 139.87, 143.88, 145.50; Anal. Calcd. For C_17_H_14_F_3_N_3_O_2_S_2_: C 49.39; H 3.41; N 10.16; %. Found: C 49.31; H 3,43; N 10.11 %.

**3-(2-(ethylthio)-4-(4-(trifluoromethyl)phenyl)-1H-imidazol-1-yl)benzenesulfonamide** (**20b**). White solid, yield 0.33 g (77%); m.p. 154–155 °C; IR (KBr) (v, cm^−1^): 3334, 1483; ^1^H NMR (400 MHz, DMSO-d_6_) δ (ppm): 1.32 (t, 3H, J = 7.3 Hz, CH_3_), 3.17 (q, 2H, J = 7.3 Hz, CH_2_), 7.58 (br. S, 2H, NH_2_), 7.70–7.85 (m, 4H, H_ar_), 7.95 (br. S, 2H, H_ar_), 8.05 (d, 2H, J = 8.1 Hz, H_ar_), 8.30 (s, 1H, CH); ^13^C NMR (101 MHz, DMSO-d_6_) (δ, ppm): 14.80, 27.49, 120.65, 122.39, 123.10, 124.73, 125.55, 125.59, 125.64, 125.81, 126.75, 127.07, 128.69, 130.53, 136.77, 137.42, 139.95, 142.74, 145.42; Anal. Calcd. For C_18_H_16_F_3_N_3_O_2_S_2_: C 50.58; H 3.77; N 9.83; %. Found: C 50.55; H 3.72; N 9.79 %.

**3-(2-(propylthio)-4-(4-(trifluoromethyl)phenyl)-1H-imidazol-1-yl)benzenesulfonamide** (**20c**). Yellowish solid, yield 0.38 g (86%); m.p. 133–134 °C; IR (KBr) (v, cm^−1^): 3352, 1486; ^1^H NMR (400 MHz, DMSO-d_6_) δ (ppm): 0.93 (t, 3H, J = 7.3 Hz, CH_3_), 1.68 (sextet, 2H, J = 7.3 Hz, CH_2_), 3.15 (t, 2H, J = 7.1 Hz, SCH_2_), 7.58 (br. S, 2H, NH_2_), 7.70–7.84 (m, 4H, H_ar_), 7.95 (br. S, 2H, H_ar_), 8.05 (d, 2H, J = 8.1 Hz, H_ar_), 8.29 (s, 1H, CH); ^13^C NMR (101 MHz, DMSO-d_6_) (δ, ppm): 13.04, 22.36, 35.08, 120.66, 122.41, 123.10, 124.71, 125.55, 125.59, 125.64, 125.80, 126.75, 127.06, 128.71, 130.52, 136.78, 137.41, 139.89, 142.87, 145.43; Anal. Calcd. For C_19_H_18_F_3_N_3_O_2_S_2_: C 51.69; H 4.11; N 9.52; %. Found: C 51.63; H 4.09; N 9.50 %.

**3-(4-(4-cyanophenyl)-2-(methylthio)-1H-imidazol-1-yl)benzenesulfonamide** (**21a**). Light yellow solid, yield 0.26 g (70%); m.p. 192–193 °C; IR (KBr) (v, cm^−1^): 3336, 1484; ^1^H NMR (400 MHz, DMSO-d_6_) δ (ppm): 2.65 (s, 3H, CH_3_), 7.58 (br. S, 2H, NH_2_), 7.77–7.89 (m, 4H, H_ar_), 7.93–8.05 (m, 4H, H_ar_), 8.34 (s, 1H, CH); ^13^C NMR (101 MHz, DMSO-d_6_) (δ, ppm): 15.36, 108.76, 119.15, 121.37, 122.19, 124.77, 125.74, 128.50, 130.65, 132.70, 136.59, 137.93, 139.63, 144.22, 145.51; Anal. Calcd. For C_17_H_14_N_4_O_2_S_2_: C 55.12; H 3.81; N 15.12; %. Found: C 55.07; H 3.78; N 15.13 %.

**3-(4-(4-cyanophenyl)-2-(ethylthio)-1H-imidazol-1-yl)benzenesulfonamide** (**21b**). Light yellow solid, yield 0.29 g (76%); m.p. 204–205 °C; IR (KBr) (v, cm^−1^): 3350, 1483; ^1^H NMR (400 MHz, DMSO-d_6_) δ (ppm): 1.32 (t, 3H, J = 7.3 Hz, CH_3_), 3.17 (q, 2H, J = 7.3 Hz, CH_2_), 7.58 (br. S, 2H, NH_2_), 7.74–8.09 (m, 8H, H_ar_), 8.34 (s, 1H, CH); ^13^C NMR (101 MHz, DMSO-d_6_) (δ, ppm): 14.80, 27.44, 108.78, 119.15, 121.37, 122.38, 124.78, 125.72, 128.71, 130.57, 132.72, 136.67, 137.95, 139.69, 143.09, 145.44; Anal. Calcd. For C_18_H_16_N_4_O_2_S_2_: C 56.23; H 4.19; N 14.57; %. Found: C 56.17; H 4.15; N 14.59 %.

**3-(4-(4-cyanophenyl)-2-(propylthio)-1H-imidazol-1-yl)benzenesulfonamide** (**21c**). White solid, yield 0.27 g (68%); m.p. 210–211 °C; IR (KBr) (v, cm^−1^): 3351, 1483; ^1^H NMR (400 MHz, DMSO-d_6_) δ (ppm): 0.93 (t, 3H, J = 7.3 Hz, CH_3_), 1.68 (sextet, 2H, J = 7.2 Hz, CH_2_), 3.15 (t, 2H, J = 7.1 Hz, SCH_2_), 7.58 (br. S, 2H, NH_2_), 7.75–8.07 (m, 8H, H_ar_), 8.33 (s, 1H, CH); ^13^C NMR (101 MHz, DMSO-d_6_) (δ, ppm): 13.05, 22.37, 35.01, 108.78, 119.15, 121.37, 122.40, 124.77, 125.73, 128.71, 130.56, 132.72, 136.68, 137.94, 139.64, 143.23, 145.44; Anal. Calcd. For C_19_H_18_N_4_O_2_S_2_: C 57.27; H 4.55; N 14.06; %. Found: C 57.21; H 4.50; N 14.08 %.

**3-(2-(methylthio)-4-(4-nitrophenyl)-1H-imidazol-1-yl)benzenesulfonamide** (**22a**). Yellow solid, yield 0.31 g (79%); m.p. 204–205 °C; IR (KBr) (v, cm^−1^): 3264, 1489; ^1^H NMR (400 MHz, DMSO-d_6_) δ (ppm): 2.66 (s, 3H, CH_3_), 7.58 (br. S, 2H, NH_2_), 7.78–7.84 (m, 2H, H_ar_), 7.97 (br. S, 2H, H_ar_), 8.09 (d, 2H, J = 8.4 Hz, H_ar_), 8.28 (d, 2H, J = 8.4 Hz, H_ar_), 8.42 (s, 1H, CH); ^13^C NMR (101 MHz, DMSO-d_6_) (δ, ppm): 15.34, 122.12, 122.19, 124.23, 124.85, 125.81, 128.51, 130.67, 136.51, 139.30, 140.00, 144.63, 145.52, 145.71; Anal. Calcd. For C_16_H_14_N_4_O_2_S_2_: C 49.22; H 3.61; N 14.35; %. Found: C 49.24; H 3.57; N 14.37 %.

**3-(2-(ethylthio)-4-(4-nitrophenyl)-1H-imidazol-1-yl)benzenesulfonamide** (**22b**). Yellow solid, yield 0.33 g (81%); m.p. 188–189 °C; IR (KBr) (v, cm^−1^): 3312, 1484; ^1^H NMR (400 MHz, DMSO-d_6_) δ (ppm): 1.33 (t, 3H, J = 7.3 Hz, CH_3_), 3.19 (q, 2H, J = 7.3 Hz, CH_2_), 7.59 (br. S, 2H, NH_2_), 7.74–7.86 (m, 2H, H_ar_), 7.96 (br. S, 2H, H_ar_), 8.09 (d, 2H, J = 8.5 Hz, H_ar_), 8.27 (d, 2H, J = 8.5 Hz, H_ar_), 8.42 (s, 1H, CH); ^13^C NMR (101 MHz, DMSO-d_6_) (δ, ppm): 14.78, 27.43, 122.08, 122.38, 124.23, 124.87, 125.79, 128.70, 130.58, 136.60, 139.37, 140.01, 143.50, 145.46, 145.73; Anal. Calcd. For C_17_H_16_N_4_O_2_S_2_: C 50.48; H 3.99; N 13.85; %. Found: C 50.49; H 3.99; N 13.79 %.

**3-(4-(4-nitrophenyl)-2-(propylthio)-1H-imidazol-1-yl)benzenesulfonamide** (**22c**). Yellow solid, yield 0.34 g (81%); m.p. 174–175 °C; IR (KBr) (v, cm^−1^): 3314, 1484; ^1^H NMR (400 MHz, DMSO-d_6_) δ (ppm): 0.94 (t, 3H, J = 7.3 Hz, CH_3_), 1.69 (sextet, 2H, J = 7.3 Hz, CH_2_), 3.17 (t, 2H, J = 7.1 Hz, SCH_2_), 7.58 (br. S, 2H, NH_2_), 7.76–7.85 (m, 2H, H_ar_), 7.96 (br. S, 2H, H_ar_), 8.08 (d, 2H, J = 8.5 Hz, H_ar_), 8.28 (d, 2H, J = 8.5 Hz, H_ar_), 8.41 (s, 1H, CH); ^13^C NMR (101 MHz, DMSO-d_6_) (δ, ppm): 13.06, 22.33, 35.00, 122.12, 122.40, 124.24, 124.86, 125.79, 128.73, 130.58, 136.61, 139.31, 140.00, 143.62, 145.45, 145.73; Anal. Calcd. For C_18_H_18_N_4_O_2_S_2_: C 51.66; H 4.34; N 13.39; %. Found: C 51.62; H 4.29; N 13.37 %.

### 2.3. Minimal Inhibitory Concentration Determination

#### 2.3.1. Preparation of Assay Microplates 

The minimal inhibitory concentrations (MICs) of compounds **2–22a–c**, as well as of clinically approved antibiotics (rifampin, isoniazid, amikacin, levofloxacin, and meropenem) were determined by microplate broth dilution method as described by Clinical Laboratory Standards Institute document M07-A8. The antimicrobials were selected to represent major antimicrobials used in clinical settings to treat MDR infections, as well as infections caused by rapidly growing *Mycobacterium* spp. The MICs for the compounds and comparator antibiotics were determined against the libraries of Gram-positive and Gram-negative pathogens, multidrug-resistant fungi, and mycobacteria. 

Compounds and antibiotics that were used as a control were dissolved in molecular biology grade dimethyl sulfoxide (DMSO) to achieve a final concentration of 25–30 mg/mL. Compound dilutions were achieved in 1.5 mL polypropylene 96-well microplates to generate 2× of concentrations of each drug (0.5–64 µg/mL). The 2× concentrates were then then transferred to flat bottom plates and used for inoculation or stored in argon purged sealed bags at −80 °C.

#### 2.3.2. Antibacterial Activity Characterization Using Gram-Positive and Gram-Negative Pathogens

A microbial inoculum was prepared using the direct colony suspension method and densitometric analysis. The inoculum suspension of each test organism was prepared in 5 mL of sterile deionized water until densitometer reached 0.5 MFa and further diluted in sterile CAMBH media to achieve final concentrations of approximately 5 × 10^5^ CFU/mL in each well after dispensing in microplates. The inoculum was transferred to the assay plates to achieve 1× assay concentration. A 10 µL of inoculum was plated on Sheep Blood agar plates to validate the purity and inoculum size. Inoculated microdilution plates were incubated at 35 °C for 16 to 20 h in an ambient-air incubator.

#### 2.3.3. Antifungal Activity Characterization

The MIC of compounds **2–22a–c**, as well as clinically approved antifungal drugs was determined by CLSI recommendations that were described in document M27-A3 [36,37]. Multidrug-resistant *Candida* spp. strains were sub-cultured on Sabouraud-Dextrose agar for 24 h at 35 °C. Drug-resistant *Aspergillus fumigatus* was cultured on Inhibitory mold agar slants for 5 days at 35 °C. The colonies of *Candida* isolates were suspended in sterile saline to reach approximately 5 × 10^6^ CFU/mL. The conidia of *A. fumigatus* were collected by flooding the slants with saline containing 0.5% of Tween 80 and passing conidia through 75 µm cell strainer. The inoculums were quantified by using haematocytometer and then, the fungal suspension was diluted in RPMI/MOPS broth to reach 5 × 10^5^ CFU/mL. The inoculum was then dispensed in assay microplates, and inoculated microdilution plates were incubated at 35 °C for 24 h in an ambient-air incubator within 15 min of the addition of the inoculum.

#### 2.3.4. Antimycobacterial Activity Determination

Before the experiments, multidrug-resistant *M. abscessus* complex strains were cultured on Middlebrook 7H9 agar containing ODAC supplement for four days at 37 °C. *M. bovis* BCG and avirulent *M. tuberculosis* H37Ra strains were grown on Lowenstein–Jensen (LJ) media for 3 weeks. 

The colonies of *M. abscessus* were scraped and suspended in tube with sterile saline to achieve approximately 5 × 10^6^ CFU/mL. *M. bovis* BCG and *M. tuberculosis* H37Ra were scraped from the LJ media and transferred to the tube containing 4 mL of Middlebrook 7H9 broth and 3 borosilicate glass beads. The tube was vortexed on maximum speed for 2 min, and then, the bacterial suspension was adjusted to 5 × 10^6^ CFU/mL. Prior to inoculation of the plates, the bacterial suspension was diluted 1:10 in Middlebrook 7H9 broth containing 20 µg/mL of resazurin, and microplates were inoculated by using multichannel pipette. 

The plates were incubated at 37 °C in humidified incubator for 5 days (for *M. abscessus* complex) or two weeks (for *M. bovis* BCG and *M. tuberculosis* H37Ra) and the minimal inhibitory concentration was determined by visual evaluation.

## 3. Results and Discussion

### 3.1. Chemistry

Most of the compounds **2**–**15** (Figure 1) were resynthesized according to our previous study [35] and were further investigated during this study. All the spectral data and reaction conditions can be found in previously mentioned research [35]. Moreover, to explore further on benzenesulfonamide-bearing 1H-imidazolethiol moieties, new compounds **3, 6, 10**, and **13** were newly synthesized for this study (Figure 1). 3-Aminobenzenesulfonamide (**1**) was treated with various α-halogenketones in water/1,4-dioxane solution to afford compound **3** and **6**. These intermediate compounds were later cyclized with potassium thiocyanate in glacial acetic acid and in a presence of HCl as a catalyst into 1H-imidazole derivatives **10** and **13**. The structures of compounds **3, 6, 10**, and **13** have also been confirmed by the data of FT-IR, ^1^H and ^13^C NMR spectroscopy, as well as elemental analysis data. For instance, in a ^1^H NMR spectrum for **10**, the singlets assigned to the protons in the CH group at 8.18 ppm and in the SH group at 13.12 ppm have proven the presence of 1H-imidazolethiol moiety in the molecule. One of the best-known properties of thioamides is the tautomerism [38]: thioamides can exist in their thione/thiol forms. However, the ^13^C NMR spectral data showed that in DMSO-d_6_ solvent, thiol tautomeric form is predominant for both compounds **10** and **13**. The carbon attributed to the C-SH group resonated at 163.17 and 163.62 ppm, respectively. 

The main goal of this study was to further investigate 1H-imidazolethiol derivatives with various alkyl substituents. For this purpose, S-alkylation reactions with bromomethane, ethyl iodide, and n-propyl iodide in dimethyl formamide were carried out to obtain compounds **16**–**22a–c**. Triethylamine was used as a base catalyst to increase the reaction rate. For example, in a ^1^H NMR spectrum for **16a**, the singlet assigned to the protons in the CH_3_ group at 2.63 ppm have proved the presence of methyl moiety in the molecule, while a triplet at 0.92 ppm, a sextet at 1.67 ppm, and a triplet at 3.12 ppm assigned to the protons in the CH_3_, CH_2_, and CH_3_, respectively, proved the presence of a propyl group in compound **16c**. Elemental analysis data of compounds **16**–**22a–c** confirmed that all the molecules did not form hydroiodide or hydrobromide salts.

### 3.2. Benzenesulfonamide Derivatives 2-22a-c Demonstrated Structure-Depended Antimicrobial Activity against Multidrug-Resistant Non-Tuberculous Mycobacteria

Novel benzenesulfonamide derivatives bearing substituted imidazoles demonstrated structure-depended antimicrobial activity against *Mycobacterium abscessus* complex strains (Table 1). Notably, compounds **2–22a–c** showed little activity against multidrug-resistant Gram-positive and Gram-negative bacterial strains or drug-resistant fungi, suggesting the mycobacteria-directed activity (Appendix A). 

Compounds **2**–**5**, bearing 4-H or halogen substitutions demonstrated no antimicrobial activity against *M. abscessus* complex strains, as well as *M. bovis* BCG or *M. tuberculosis* H37Ra (MIC > 64 µg/mL). The addition of the 4-CF_3_ substitution on the benzenesulfonamide core in compound **6** resulted in weak antimicrobial activity against *M. abscessus* complex strains (MIC 64 µg/mL) except for *M. abscessus* MA1836. Moreover, compound **7** showed no activity against *M. bovis* BCG or *M. tuberculosis* H37Ra strains (MIC > 64 µg/mL). Furthermore, the incorporation of 4-CN (**7**), or 4-NO_2_ (**8**), in the benzenesulfonamide nucleus diminished the antimicrobial activity against *M. abscessus* complex, as well as *M. bovis* BCG or *M. tuberculosis* H37Ra (Table 1).

The incorporation of imidazole-2-thiol moiety in compound **9** resulted in weak antimicrobial activity against *M. abscessus* complex strains (MIC 64 µg/mL) with exception of *M. abscessus* MA1704 and MA0040 (MIC > 64 µg/mL). The incorporation of imidazole-2-thiol moiety (compound **9**) resulted in extended antimicrobial activity against rapidly growing *M. abscessus* strains, non-tuberculous mycobacteria (*M. bovis* BCG), as well as *M. tuberculosis* H37Ra (MIC 32 µg/mL, respectively). Interestingly, 4-Br, 4-Cl substitutions in imidazole-2-thiol derivatives (**10,11**) resulted in loss of antimicrobial activity against mycobacteria, while 4-F substitution (**12**) resulted in antimicrobial activity against *M. abscessus* complex (MIC 32–64 µg/mL) and loss of activity against *M. bovis* BCG and *M. tuberculosis* H37Ra (MIC > 64 µg/mL). The further addition of 4-CF_3_ substitution resulted in compound **13** with strong antimicrobial activity against tested mycobacterial strains (MIC 0.5–4µ g/mL). The antimicrobial activity of compound **13** against *M. abscessus* complex was greater than rifampicin (MIC 32–64 µg/mL), isoniazid (MIC 4–32 µg/mL), amikacin (MIC 16–32 µg/mL), levofloxacin (MIC 8–32 µg/mL), and meropenem (MIC 8–64 µg/mL) (Table 1). 

The incorporation of an aryl group often results in increased lipophilicity of the compounds. Therefore, we further postulated that the incorporation of various length aryl substitutions if benzenesulfonamide derivatives could enhance the mycobacteria-directed antimicrobial activity. Compound **16a** bearing methyl group demonstrated weak antimicrobial activity against *M. abscessus* complex strains MA1884 and MA1753 (MIC 64 µg/mL). The elongation of the aryl chain by adding ethyl and propyl groups (**16b** and **16c**) diminished the antimicrobial activity. On the other hand, compounds **18a–c** containing the 4-Cl substitution demonstrated that the length of the aryl chain is mediating the antimicrobial activity. Compound **18a** bearing the methyl substitution showed no antimicrobial activity while compound **18b** containing the ethyl group showed antimicrobial activity against *M. abscessus* complex (MIC 32–64 µg/mL), but not *M. bovis* BCG or *M. tuberculosis* H37Ra (MIC > 64 µg/mL). Notably, the incorporation of propyl substitution (**18c**) resulted in enhanced antimicrobial activity against all tested *M. abscessus* complex strains (MIC 16–64 µg/mL), as well as *M. bovis* BCG (MIC 32 µg/mL) and *M. tuberculosis* H37Ra (MIC 16 µg/mL). Furthermore, compound bearing 4-F substituent and methyl group (**19a**) showed good antimicrobial activity against *M. abscessus* complex strains, as well as *M. bovis* BCG and *M. tuberculosis* H37Ra (MIC 4–8 µg/mL respectively). However, other S-alkyl groups–ethyl (**19b**) and propyl (**19c**) in imidazole bearing 4-fluorophenyl substituent completely diminished antimicrobial activity against tested strains (Table 1). 

## 4. Conclusions

During this study, a series of imidazole-2-thiol bearing benzenesulfonamides was synthesized. To reach higher lipophilicity properties and potentially increase their membrane permeability through multidrug-resistant mycobacteria, various S-alkylation reactions were performed with alkyl halides. 

Synthesized compounds showed structure-dependent antimicrobial activity against *Mycobacterium abscessus* complex strains. Furthermore, compounds **2–22a–c** showed little activity against multidrug-resistant Gram-positive and Gram-negative bacterial strains or drug-resistant fungi. However, 3-(2-thioxo-4-(4-(trifluoromethyl)phenyl)-2,3-dihydro-1*H*-imidazol-1-yl)benzenesulfonamide (**13**) has demonstrated high antibacterial activity against all tested mycobacterial strains and was more active than widely used antibiotics like rifampin, amikacin, or levofloxacin. 

Previous studies have explored the impact of alkyl substitution on the antimicrobial activity of various compounds against mycobacteria and other clinically important pathogens. Oh et al. [39] have reported the synthesis of a series of novel *N*-Alkyl-5-hydroxypyrimidinone carboxamides as potent inhibitors of *M. tuberculosis* decaprenylphosphoryl-β-d-ribose 2’-oxidase. Faria et al. [40] describes alkyl promising activity and the high reactivity of alkyl hydrazide derivatives of isoniazid, suggesting that the alkylation is an important modification leading to the in vitro and in silico activity. Yang Yong et al. [41] described the synthesis of novel 8-alkylberberine derivatives bearing aliphatic chains and evaluated their antimicrobial activity. The study showed that increasing the length of the aliphatic chain had a significant effect on the antibacterial activity of the compounds. However, antimicrobial activity started to decrease when alkyl chain consisted eight or more carbon atoms. 

S-alkylation is widely employed strategy to increase the stability of biologically active compounds due to higher bond dissociation energy of the S-C bond compared to the N-C bond [42,43]. S-alkylated compounds are generally less susceptible to hydrolysis and more resistant to metabolic degradation compared to N-alkylated compounds, making S-alkylation an attractive strategy to enhance the biological activity of various compounds.

In our study, we compared S-alkylated benzenesulfonamide bearing imidazole derivatives against multidrug-resistant *M. abscesus* complex strains, and we found that 3-(4-(4-fluorophenyl)-2-(methylthio)-1*H*-imidazol-1-yl)benzenesulfonamide (**19a**) and 3-(4-(4-chlorophenyl)-2-(propylthio)-1*H*-imidazol-1-yl)benzenesulfonamide **18c** showed the highest antimycobacterial activity. For instance, MICs of compound **19a** with 4-fluorophenyl and S-methyl substituents against *M. abscessus* complex strains, as well as *M. bovis* BCG and *M. tuberculosis* H37Ra, were 4–8 µg/mL, respectively. However, ethyl or propyl groups in the same 1*H*-imidazol-2-thiol scaffold with 4-fluorophenyl group (compounds **19b** and **19c**) reduced the potency significantly. As for the imidazole scaffold with 4-chlorophenyl substituent, antimicrobial activity was increased by extending the alkyl chain. Compound **18c** containing S-propyl group was more potent than **18a** (S-methyl) and **18b** (S-ethyl) compounds.

These results suggest that the S-alkylated benzenesulfonamide-bearing imidazole derivatives could be further explored as a scaffold for the development of novel, multidrug-resistant *M. abscesus* complex-directed antimicrobials. 

## Data Availability

Data is contained within the article and Appendix A. The compounds are available from the corresponding author.

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
