# Peer review of "Synthesis of Novel Benzenesulfonamide-Bearing Functionalized Imidazole Derivatives as Novel Candidates Targeting Multidrug-Resistant Mycobacterium abscessus Complex"

_microorganisms, 2023, doi:10.3390/microorganisms11040935_

Round 1

Reviewer 1 Report

The manuscript by Balandis et al. is well written and conducted, adding interesting data regarding novel drug candidates targeting MDR Mycobacterium abscessus complex. I encourage its further processing after appropriate minor modifications as outlined below:

Line 22: are an important public health concern (the  word "an " is missing)

Line 47:  "develop novel " without  "a "

Line 57: I think it should be  "parenteral " not parental

Line 74: reported to be  "a promising antimicrobial target

Line 75: the correct form is  "leads ", not  "lead "

Line 87-88: the word  "bearing " is repeated twice in the same sentence. I would suggest rephrasing the sentence.

Line 295-296:  " as well as of clinically approved antibiotics "

Line 300:  "the " CLSI document

Line 318: the comma after Candida spp. is not necessary

Line 343:  "previous " instead of  "previously "

Other comments:

-a more extensive discussion section would have added value to the manuscript (the author might try to compare the results of their study with other previously reported results)

-MIC determination: explain why the mentioned antibiotics were chosen for this determination 

Author Response

Response to Reviewer 1 Comments

Point 1: More extensive discussion section would have added value to the manuscript (the author might try to compare the results of their study with other previously reported results).

Response 1: Thank you for the comment. We agree that discussion section needed some improvements. Mycobacterium as well as M. abscesus have lipid bilayer which can limit molecule transfer into the cell. For this reason, we have performed S-alkylation reactions to increase the lipophilicity of the molecules. To improve aforementioned section, we expanded the discussion by describing other study results to further explore the benefits of incorporating alkyl chains into the molecules.

Point 2: MIC determination: explain why the mentioned antibiotics were chosen for this determination.

Response 2: Thank you for great question. The comparative antibiotics were chosen based on the following criteria:

  1. The antibiotics are used alone or in combinations with other drugs to treat infections caused by M. abscesus complex bacteria in the clinical setting.
  2. The antibiotics represents several classes of antimicrobials.
  3. The antibiotics are active against rapidly growing and tuberculiosis-type of Mycobacterium.

We are incorporating the rationale on selection of comparative antibiotics to the manuscript.

Other mistakes in text were corrected in a new version of the manuscript according to your detailed comments.

Reviewer 2 Report

In the present manuscript, the authors designed and synthesized benzenesulfonamide compounds with imidazole and investigated their antimicrobial activity against various multidrug-resistant strains with emphasis on M. abscessus. Overall, the manuscript is well written, structured and presented and addresses an important issue, antibacterial resistance and the urgent need for new antibacterial compounds.

Here are my comments.

1.       General information on manufacturers of chemicals and equipment for the determination of compounds should be given in the Materials and Methods section, since some compounds were synthesized and characterized for the first time in this manuscript.

2.       Scheme 1 should better indicate which compound has which substituent. In the current version, the substituents for compounds 9-15 and 16-22a-c are only listed and not assigned to the compounds, so it is difficult to determine which compound is which. Please clearly indicate which compound has which substituent. Since the compounds are not properly labelled, it is difficult to interpret the results.

3.       I believe there is an error in Table 1 as well in Tables S1 and S2. Did the authors accidentally write <64 instead of >64? The current presentation of the results shows that the majority of compounds have a MIC of less than 64 (MIC<64). Reading the discussion of the results, one understands that compounds with MIC<64 are worse than compounds with MIC=64, which makes no sense. If the sign is rotated incorrectly, please correct, otherwise please provide the lowest MIC value determined. All Tables are missing units and a number of repetitions.

4.       Lines 394-396: "Interestingly, the incorporation of imidazole-2-thiol moiety resulted in a broad anti-mycobacterial spectrum, since compound 9 showed antimicrobial activity against M. bovis BCG and M. tuberculosis H37Ra (MIC 32 μg/ml respectively)." In my opinion, the phrase "broad antimycobacterial spectrum" is exaggerated. The antibacterial activity is well above that determined for known antibiotics.

5.       Since the authors have incorporated alkyl chains to increase lipophilicity and improve penetration, I miss the discussion of how alkyl-substituted compounds affect MIC compared to their unsubstituted analogs.

6.       The Conclusion shows some compounds that have not been assigned a number. Please add the number of the compound.

Author Response

Response to Reviewer 2 Comments

Point 1: General information on manufacturers of chemicals and equipment for the determination of compounds should be given in the Materials and Methods section, since some compounds were synthesized and characterized for the first time in this manuscript.

Response 1: Thank you for great comment. Materials and Methods section was updated accordingly in a new revision of the manuscript.

Point 2: Scheme 1 should better indicate which compound has which substituent. In the current version, the substituents for compounds 9-15 and 16-22a-c are only listed and not assigned to the compounds, so it is difficult to determine which compound is which. Please clearly indicate which compound has which substituent. Since the compounds are not properly labelled, it is difficult to interpret the results.

Response 2: Thank you for the good remark. We agree and Scheme 1 was updated according to your comment.

Point 3. I believe there is an error in Table 1 as well in Tables S1 and S2. Did the authors accidentally write <64 instead of >64? The current presentation of the results shows that the majority of compounds have a MIC of less than 64 (MIC<64). Reading the discussion of the results, one understands that compounds with MIC<64 are worse than compounds with MIC=64, which makes no sense. If the sign is rotated incorrectly, please correct, otherwise please provide the lowest MIC value determined. All Tables are missing units and a number of repetitions.

Response 3: Thank you for the observation. We apologize for the topographical mistake, and we are incorporating the changes in the manuscript as well as supplementary files.

Point 4. Lines 394-396: "Interestingly, the incorporation of imidazole-2-thiol moiety resulted in a broad antimycobacterial spectrum, since compound 9 showed antimicrobial activity against M. bovis BCG and M. tuberculosis H37Ra (MIC 32 μg/ml respectively)." In my opinion, the phrase "broad antimycobacterial spectrum" is exaggerated. The antibacterial activity is well above that determined for known antibiotics.

Response 4: Thank you for the comment. We agree with the suggestion and incorporating the changes.

Point 5. Since the authors have incorporated alkyl chains to increase lipophilicity and improve penetration, I miss the discussion of how alkyl-substituted compounds affect MIC compared to their unsubstituted analogs.

Response 5: We are thankful for the comment. We are providing the paragraph on the N and  S-alkylation, rationale in choosing S-alkylation and literature as well as our data comparisons.

Point 6. The Conclusion shows some compounds that have not been assigned a number. Please add the number of the compound.

Response 6: Thank you for great observation. Number of the compound was added in the conclusion section.

Reviewer 3 Report

This paper describes the synthesis and biological evaluation of some imidazole compounds containing benzenesulfonamide moieties to enhance membrane penetration.  The chemistry follows that reported previously by this group, but makes a significant further development, with a large number of new compounds, which have all been well-characterised.  The biological results add to knowledge in this field, and one compound shows strong antimicrobial activity.  I recommend the paper for publication, but the English language needs to be checked and corrected (mainly with respect to the use of singular and plural) prior to acceptance.

Author Response

Thank you for your extensive review. We apologize for many English language mistakes which were left in the text. We checked and corrected them, mainly focusing on the use of plural/singular.

line 340 to 344 :Drug-resistant Aspergillus fumigatus was cultured on Inhibitory mould agar slants for 5 days at 35°C. The colonies of Candida isolates were suspended in sterile saline to reach approximately 5 × 106 CFU/ml. The conidia of A. fumigatus were collected by flooding the slants with saline containing 0.5% of Tween 80 and passing conidia through 75 µm cell strainer.